# Identification and Genomic Characterization of *Escherichia albertii* in Migratory Birds from Poyang Lake, China

**DOI:** 10.3390/pathogens12010009

**Published:** 2022-12-21

**Authors:** Qian Liu, Xiangning Bai, Xi Yang, Guoyin Fan, Kui Wu, Wentao Song, Hui Sun, Shengen Chen, Haiying Chen, Yanwen Xiong

**Affiliations:** 1State Key Laboratory of Infectious Disease Prevention and Control, National Institute for Communicable Disease Control and Prevention, Chinese Center for Disease Control and Prevention, Beijing 102206, China; 2Division of Clinical Microbiology, Department of Laboratory Medicine, Karolinska Institute, 141 52 Stockholm, Sweden; 3Nanchang Center for Disease Control and Prevention, Nanchang 330038, China

**Keywords:** *Escherichia albertii*, genomic analysis, migratory birds, Poyang Lake

## Abstract

*Escherichia albertii* is an emerging zoonotic foodborne enteropathogen leading to human gastroenteritis outbreaks. Although *E. albertii* has been isolated from birds which have been considered as the potential reservoirs of this bacterium, its prevalence in migratory birds has rarely been described. In this study, *E. albertii* in migratory birds from Poyang Lake was investigated and characterized using whole genome sequencing. Eighty-one fecal samples from nine species of migratory birds were collected and 24/81 (29.6%) tested PCR-positive for *E. albertii*-specific genes. A total of 47 isolates was recovered from 18 out of 24 PCR-positive samples. All isolates carried *eae* and *cdtB* genes. These isolates were classified into eight *E. albertii* O-genotypes (EAOgs) (including three novel EAOgs) and three *E. albertii* H-genotypes (EAHgs). Whole genome phylogeny separated migratory bird-derived isolates into different lineages, some isolates in this study were phylogenetically closely grouped with poultry-derived or patient-derived strains. Our findings showed that migratory birds may serve as an important reservoir for heterogeneous *E. albertii*, thereby acting as potential transmission vehicles of *E. albertii* to humans.

## 1. Introduction

*Escherichia albertii* is a newly described *Escherichia* species and emerging foodborne pathogen causing watery diarrhea, abdominal distention, vomiting, fever, and even bacteremia in humans [1]. *E. albertii* was first identified in diarrheal children from Bangladesh and has been associated with several human gastroenteritis outbreaks in Japan [2,3]. Due to the lack of distinguishing biochemical characteristics, *E. albertii* strains were often misidentified as *E. coli*, *Hafnia alvei*, *Salmonella enterica*, or *Shigella boydii* serotype 13 [4]. Thus, the prevalence of *E. albertii* in different hosts may well have been underestimated.

*E. albertii* possesses an outer membrane protein intimin encoded by *eae* gene, which is responsible for the formation of attaching effacing (A/E) lesions on host intestinal epithelium [1]. In addition, almost all *E. albertii* isolates harbor a *cdtABC* locus which encodes the cytolethal distending toxin (CDT) [5]. The *cdtB* gene has been divided into five subtypes (*cdtB*-I to *cdtB*-V) in *E. coli*. A new subtype, *cdtB*-VI, was recently reported in *E. albertii* [6]. Shiga toxins (*stx2a* and *stx2f*) were identified in some *E. albertii* isolates [7,8]. Other virulence factors reported in *E. albertii*, such as enteroaggregative *E. coli* heat-stable enterotoxin encoded by *astA*, may contribute to pathogenicity, but they have not been systematically investigated [6].

*E. albertii* has been isolated from various animal sources, such as poultry, pigs, cats, dogs, bats, and raccoons as well as animal-derived raw meats [9,10], yet its natural reservoirs and transmission route to humans remain uncertain. Previous epidemiological investigations demonstrated that birds may be considered as potential reservoirs of *E. albertii* [11,12,13]. Migratory birds can travel over long distances and carry pathogens from one region to another, spreading pathogens by feeding and excretion [14,15]. Poyang Lake, a globally important wintering and transfer point for migratory birds in the East Asian–Australasian Flyway (EAAF), harbors more than 400,000 birds belonging to around 87 species in the winter season [16]. Cranes, geese, and swans are the most abundant migratory birds found in Poyang Lake [17]. Every year, they start their northward migration at Poyang Lake and progress through inland China, Korea, Japan, or Russia from March to May [18].

In this study, for the first time, we reported the identification of *E. albertii* in migratory birds from Poyang Lake, China. Genomic characterization indicated the genetic diversity of migratory bird-derived *E. albertii* isolates, and some isolates may have the potential to cause human disease.

## 2. Materials and Methods

### 2.1. Sample Collection

The stool samples were collected from migratory birds at two different sites (116°11′35.178″ N, 29°15′15.422″ E and 116°18′38.912″ N, 29°9′57.287″ E) in Poyang Lake, China. All samples in this study were collected in March 2020 when most migratory birds were migrating to their breeding areas. To ensure the diversity of samples, different species of migratory birds were captured. The species of migratory birds were identified by the accompanied ornithologists. After live capture, birds were separately placed in the grill cages for 5–10 min until they defecated. A sterile tray was placed under each grill cage to prevent feces from being contaminated. Once the bird defecated, the tray was extracted, and the stool sample was rapidly transferred to sterile 2 mL centrifuge tubes. Then, the samples were placed on dry ice, transported to the laboratory, and stored at −80°C until further analysis. The grill cages and the trays were entirely sterilized before and after the collection phase with a 10% spray bleach solution (Topchemical, Shenzhen, China) for at least 10 min to reduce environmental and cross-contamination. All migratory birds were seemingly healthy and released safely after sample collection. The animal welfare practices associated with this study were reviewed and approved by the Jiangxi Provincial Department of Forestry, China (No. 20181030).

### 2.2. Isolation and Identification of E. albertii

The stool samples were enriched with *Escherichia coli* broth (Land Bridge, Beijing, China) and incubated at 20 °C for 24–36 h on a shaking platform (220 rpm). Enriched samples were examined by PCR for the presence of *EAKF1_ch4033* gene described by Lindsey et al. [19]. To describe the diversity of *E. albertii* carried by the same individual, each PCR-positive enrichment was streaked onto 4 MacConkey agar (Oxoid, Hampshire, UK) plates in parallel. After 20–24 h of incubation at 37 °C, colorless colonies (maximum 12 colonies per plate) were picked and tested for *EAKF1_ch4033* by single colony PCR assay. *EAKF1_ch4033*-positive colonies were determined to be *E. albertii* by diagnostic PCR targeting *clpX, lysP,* and *mdh* genes, as previously reported [20]. Finally, only one PCR-positive isolate from each plate (~4 isolates per sample) was retained for further analysis.

### 2.3. Antimicrobial Susceptibility Test

The antimicrobial susceptibility tests were determined using the BD Phoenix^TM^ M50 Automated Microbiology System (BD, San Jose, CA, USA) as previously described [21]. Mueller Hinton broth was used as the test medium. The Gram-negative panel RUONMIC-801 was selected in this study. The dilution ranges (μg/mL) for the seventeen antimicrobial agents used in the panels were as follows: amikacin (AMK, 4–64 μg/mL), ampicillin (AMP, 2–32 μg/mL), ampicillin-sulbactam (SAM, 1/0.5–32/16 μg/mL), azithromycin (AZM, 2–64 μg/mL), aztreonam (ATM, 0.25–16 μg/mL), cefotaxime (CTX, 0.25–16 μg/mL), cefoxitin (FOX, 2–64 μg/mL), ceftazidime (CAZ, 0.25–16 μg/mL), chloramphenicol (CHL, 4–32 μg/mL), ciprofloxacin (CIP, 0.015–2 μg/mL), ertapenem (ETP, 0.25–8 μg/mL), imipenem (IPM, 0.25–8 μg/mL), meropenem (MEM, 0.125–8 μg/mL), nalidixic acid (NA, 4–32 μg/mL), nitrofurantoin (FM, 32–256 μg/mL), tetracycline (TE, 1–16 μg/mL), trimethoprim–sulfamethoxazole (SXT, 0.5–8 μg/mL). Bacterial suspension and panel inoculation were performed according to the manufacturer’s guidelines. Panels were incubated for 16–20 h at 35°C, and results were interpreted with BD EpiCenter software. *Escherichia coli* ATCC 25,922 was used for quality control. Clinical Laboratory Standards Institute guidelines (CLSI) breakpoints for *Enterobacterales* were used to define the isolates as susceptible (S), intermediate (I), or resistant (R) [22].

### 2.4. Whole Genome Sequencing and Assembling

Genomic DNA of all isolates was extracted from overnight culture using the Wizard Genomic DNA purification kit (Promega, Madison, WI, USA) according to the manufacturer’s instructions. Bacterial genomes were sequenced and assembled as previously described [23]. One representative *E. albertii* isolate from each sample was sequenced using the combined methods of the PacBio Sequel (Pacific Biosciences, Menlo Park, CA, USA) and Illumina NovaSeq 6000 platform (Illumina, San Diego, CA, USA) to obtain complete genomes. The raw PacBio sequencing reads were first quality-controlled with the “RUN QC” module in SMRT Link version 5.1.0 (www.pacb.com/support/software-downloads accessed on 18 September 2022) and de novo assembled using the hierarchical genome assembly process (HGAP) pipeline [24], then corrected with the Illumina short reads. The remaining isolates were sequenced using the Illumina platform as described above. The paired-end reads were filtered by fastp v0.20.1 (https://github.com/OpenGene/fastp accessed on 18 September 2022) [25] and assembled using SKESA v2.4.0 [26].

### 2.5. Molecular Characterization of E. albertii Isolates

The *E. albertii* H-genotypes (EAHgs) were determined using BLAST + search against four H-genotypes sequences described by Nakae et al. with a coverage ≥ 90% and identity ≥ 99% [27]. The *E. albertii* O-genotypes (EAOgs) were determined using BLAST+ search against 42 primer pairs described by Ooka et al. [28]. For unmatched isolates, each O-antigen biosynthesis gene cluster (O-AGC) between *galF* and *gnd* genes was extracted from the genome sequence. Open reading frames (ORFs) were predicted using Prokka v1.14.6 [29]. Functional annotation of the ORFs was performed based on the results of a homology search against the public, non-redundant protein database using BLASTP. The DeepTMHMM v1.0.10 analysis program (https://dtu.biolib.com/DeepTMHMM accessed on 10 October 2022) [30] was used to identify potential transmembrane segments from the amino acid sequences. The genetic structures of O-AGC were visualized using EasyFig v2.2.5 [31].

To determine the presence and subtypes of *eae* and *cdtB* genes, all *E. albertii* genomes in this study were compared to the representative sequences of all described *eae* and *cdtB* subtypes using BLAST + with an identity ≥ 95% and coverage ≥ 95% [13]. Virulence and antimicrobial resistance (AMR) genes were identified by comparing *E. albertii* genomes against the reference sequences in VirulenceFinder database (https://cge.food.dtu.dk/services/VirulenceFinder/ accessed on 10 October 2022) and ResFinder database (https://cge.food.dtu.dk/services/ResFinder/ accessed on 10 October 2022), using ABRicate v1.0.1 (https://github.com/tseemann/abricate accessed on 10 October 2022) with default parameters [32,33].

### 2.6. Phylogenomic Analysis

To elucidate the population structure of *E. albertii* isolated from migratory birds in Poyang Lake, representative *E. albertii* genomes from different sources and countries were downloaded from NCBI (https://www.ncbi.nlm.nih.gov/ accessed on 15 October 2022). QUAST v4.5 was used to evaluate the downloaded genomes [34]. Genome contamination and completeness were evaluated utilizing the CheckM taxonomic-specific (species) workflow [35]. Single nucleotide polymorphisms (SNPs) were identified by mapping the *E. albertii* genomes to the reference genome, *E. albertii* CB9786 (GCA_002285475.1) by using Snippy v4.6.0 (https://github.com/tseemann/snippy accessed on 15 October 2022) with the default parameters. Gubbins v3.1.6 [36] was used to concatenate SNPs and remove recombination regions with default parameters. FastTree v2.1.11 [37] was used to reconstruct the phylogeny and generalize time-reversible nucleotide substitution with gamma correction model. Core genome SNP (cgSNP) distances were calculated using snp-dists v0.8.2 (https://github.com/tseemann/snp-dists accessed on 15 October 2022). Fastbaps v1.0.4 was used to identify the lineages of *E. albertii* as previously described [6,38]. The phylogenetic tree was visualized using Chiplot (https://www.chiplot.online/ accessed on 16 October 2022).

### 2.7. Data Availability

All genomes in this work were submitted to GenBank under the accession numbers CP099868–CP099914 and JAMXMZ000000000–JAMXOB000000000. The annotated se-quences of O-antigen biosynthesis gene clusters were submitted to GenBank under accession numbers OP019328-OP019330.

## 3. Results

### 3.1. Prevalence of E. albertii in Fecal Samples of Different Migratory Bird Species

A total of 81 fecal samples from nine species of migratory birds were collected. The major species from which the samples were collected were Eurasian wigeon (*Mareca penelope*) (*n* = 31), Taiga bean goose (*Anser fabalis*) (*n* = 19), Greater white-fronted goose (*Anser albifrons*) (*n* = 11), and Northern pintail (*Anas acuta*) (*n* = 11). Of the 81 samples, 24 (29.6%) yielded a 393-bp PCR amplicon specific for *E. albertii*, and 18 (22.2%) PCR-positive samples were culture-positive for *E. albertii* (Table 1). Isolates were recovered from six species of migratory birds, including Eurasian wigeon, Taiga bean goose, Greater white-fronted goose, Northern pintail, Lesser white-fronted goose, and Tundra swan. A single isolate was obtained from three fecal samples, two isolates per sample were recovered from four samples each, three isolates per sample were obtained from eight samples each, and four isolates each were obtained from three samples. A total of 47 *E. albertii* isolates were kept for further analysis (Table 1).

### 3.2. Antimicrobial Susceptibility and Antimicrobial Resistance (AMR) Genes

Twelve isolates (25.5%) showed resistance to tetracycline; all isolates, except one, carried tetracycline-associated resistant gene *tetB*. All isolates possessed macrolide-associated resistance gene *mdfA*, while only six isolates showed resistance to azithromycin. Twelve isolates possessed sulfonamide-associated resistance gene *sul2*, while all of them were susceptible to trimethoprim–sulfamethoxazole. All 47 isolates were susceptible to the other 15 antimicrobials tested (Appendix A).

### 3.3. E. albertii O/H-Antigen Genotypes

Three *E. albertii* H-genotypes (EAHgs), i.e., EAHg1, EAHg3, and EAHg4 were identified from all isolates. The major EAHgs were EAHg4 (24 isolates, 51.0%), followed by EAHg3 (17 isolates, 36.2%) and EAHg1 (6 isolates, 12.8%) (Table 2). Five previously reported *E. albertii* O-genotypes (EAOgs), i.e., EAOg1, EAOg2, EAOg6, EAOg8, and EAOg21 [28,39,40,41,42,43] were identified from 38 isolates. The major EAOgs were EAOg2 and EAOg21, including 16 (34.0%) and 14 (29.8%) isolates, respectively (Table 2). The remaining nine isolates cannot be assigned to any known EAOgs. According to the O-AGCs, three novel EAOgs, i.e., EAOg41, EAOg42, and EAOg43 were proposed. The size of these novel O-AGCs between *galF* and *gnd* genes ranged from 13.2 kb to 14.5 kb, and the G + C content ranged from 37.0 to 38.0 mol%, within the values previously reported [28,40]. The O-AGCs of EAOg41 and EAOg42 were similar to partial O-AGCs of *E. coli* and *E. albertii*, with nucleotide sequence similarity ranging from 60% to 100%. However, the entire O-AGC of EAOg43 showed resemblance to *E. coli* O131 counterparts, showing nearly 100% nucleotide sequence identity (Figure 1).

### 3.4. The eae, cdtB, and Other Virulence Genes

All 47 *E. albertii* isolates carried *eae* gene belonging to six subtypes. The most common subtypes were sigma and epsilon4, accounting for 40.4% (19/47) and 29.8% (14/47), respectively. Other subtypes were epsilon3 (6 isolates), epsilon1 (3 isolates), nu (3 isolates), and N3 (2 isolates). All isolates harbored *cdtB* belonging to two subtypes *cdtB*-II (28 isolates) and *cdtB*-VI (19 isolates). The virulence-related genes *tir*, *chuA*, *espA*, *espF*, *gad*, and *terC* were present in all isolates, while the toxin genes *astA*, *cma*, and *cmb* were identified in 31.9% (15/47), 70.2% (33/47), and 70.2% (33/47) isolates, respectively (Appendix A).

### 3.5. Genomic Variations among 47 E. alberii Isolates

The genomic size of 18 complete *E. alberii* genomes ranged from 4,657,64 to 5,122,791 bp, with gene number ranging from 4477 to 4942. Plasmids were identified in 15 out of 18 complete genomes. Among these 15 isolates, three possessed only one plasmid, eleven isolates possessed two plasmids, and one isolate possessed four plasmids. The genome sizes of the 29 draft genomes ranged from 4,575,346 to 5,334,983 bp, with gene number ranging from 4461 to 5228. The GC content of all isolates was approximately 49% (Appendix A).

A maximum-likelihood phylogenetic tree based on 83,912 SNPs identified among 47 *E. albertii* isolates was constructed. Six main clades were observed in the phylogenetic tree (Figure 2). Three samples (sample ID: 121, 153, and 253) recovered 2–3 isolates each. These isolates separated into different clades. For example, isolates 121_1_EW_A and 121_2_EW_A were separated into Clade 2 and Clade 4.2, respectively. Isolates 153_1_TBG_A, 153_2_TBG_A, and 153_3_TBG_A were divided into Clade 3, Clade 4.2, and Clade 2. Isolates 253_1_EW_B, 253_2_EW_B, and 253_3_EW_B were grouped into Clade 3, Clade 4.1, and Clade 4.2. This may indicate that various clones of *E. albertii* coexisted in the same individual. Notably, some isolates from different migratory birds showed highly phylogenetical relatedness (e.g., Clade 3 and Clade 4.2), although the migratory birds were generally captured in different sites around Poyang Lake (Figure 2A).

### 3.6. Global Comparative Analysis of E. albertii

Among the 15 samples recovered with more than one isolate each, eight samples (e.g., 105, 121, 153, 205, 208, 222, 251, and 253) yielded different isolates based on their molecular characteristics and SNPs. Some isolates derived from the other seven samples (e.g., 104, 110, 133, 147, 155, 233, and 247) showed similar antimicrobial phenotypes and molecular characteristics, and were clustered into the same clade in the phylogenetic tree (Appendix A, and Figure 2A). This may indicate that these isolates are derived from the same clone. One of the clonal strains from same samples was used to understand the genetic relationship of *E. albertii* populations from birds, humans, and other sources. A phylogenetic tree was constructed based on 164 *E. albertii* genome sequences downloaded from NCBI and Entrobase database (Appendix A), and 29 genome sequences from this study. A previous study showed that *E. albertii* strains were divided into eight lineages (L1–L8) [6]. The phylogenetic tree showed that 22 isolates from migratory birds fell into three different lineages (L5, L7, and L9), while the remaining seven isolates did not belong to any lineages (Figure 2B). In the three lineages, L5 and L7 were proposed in the previous study [6], and L9 was expanded and first defined in this study. Some isolates were phylogenetically related to poultry- or patient-derived strains, especially in lineage L7 (Figure 2B).

To further define genomic distance between migratory bird-derived isolates and phylogenetically related strains, the pairwise cgSNP (core genome SNP) values were calculated and analyzed (Figure 3 and Appendix A). In L5, the cgSNPs between three migratory bird-derived isolates and phylogenetically related strains differed from 755 to 1096 cgSNPs. In L7, the cgSNPs between nineteen migratory bird-derived isolates and phylogenetically related strains (including one strain SRR1999986 out of lineages) differed from 97 to 8267 cgSNPs. In L9, the cgSNPs between fourteen migratory bird-derived isolates and phylogenetically related strains differed from 458 to 7787 cgSNPs (Appendix A). The strains showing phylogenetic relatedness to the migratory bird isolates were mainly isolated from human (13 strains from UK, Japan, Guinea, Poland, and China), poultry (9 strains from China), bird (2 strains from Japan and France), dog (1 strain from Australia), and unknown source (1 strain from USA) from 1994 to 2019 (Figure 3). Thirteen human-derived strains were collected from clinical settings and one of them (GCA_001515065.1) was identified as a causative bacterium of a human gastroenteritis outbreak in Japan in 2011 [3]. Nine phylogenetically related poultry-derived strains were collected in different provinces of China. Notably, one isolate 105_3_LWG_A in L7 showed closely phylogenetic relationship with a poultry-derived strain (SRR13494886) and a human-derived strain (SRR12769693), with an average of 101 cgSNPs. Moreover, these two strains were collected from China (SRR13494886) in 2014 and UK (SRR12769693) in 2019, respectively (Figure 3).

## 4. Discussion

*E. albertii* is known to be an emerging zoonotic foodborne pathogen and has been isolated from several species of wild birds (Redpoll finches, European wigeon, Pine siskins, Magpies, Pigeons, and others) worldwide, demonstrating its diverse reservoirs and global distribution [12,44]. In the present study, for the first time, we reported *E. albertii* in different migratory birds in Poyang Lake, China. Besides Eurasian wigeon, *E. albertii* were first identified from Taiga bean goose, Greater white-fronted goose, Northern pintail, Lesser white-fronted goose, and Tundra swan. The overall culture-positive rate was 22.2% (18/81) in this study, which was higher than that reported in birds from other countries (0.7–3.2%) [12,44].

Serotyping plays an important role in diagnosis and epidemiological studies for pathogens of public health importance. For example, most reported outbreaks of *E. coli* have been attributed to several serogroups (e.g., O26, O111, and O157) [45]. The diversity of O-antigen biosynthesis gene clusters (O-AGCs) provides the primary basis for serotyping. Instead of the conventional agglutination test, forty O-genotypes (named EAOg1–EAOg40) and four H-genotypes (EAHg1-EAHg4) unique to *E. albertii* have been proposed [27,28]. The O-antigen genotypes of *E. albertii* were associated with virulence genes. For example, the EAOg18 strains were predominant in human-derived strains and often harbored *stx2f* gene [46]. In this study, 38 isolates belonged to five known EAOgs (EAOg1, EAOg2, EAOg6, EAOg8, and EAOg21) and three EAHgs (EAHg1, EAHg3, and EAHg4). Three novel O-AGCs, named as EAOg41–43, were identified among nine isolates in this study, indicating the high diversity of *E. albertii* in migratory birds.

The *eae* and *cdtB* genes were commonly considered as the key virulence determinants of *E. albertii* [5]. Currently, at least 30 *eae* subtypes have been described in *E. coli*. Some subtypes such as beta1, gamma1, and sigma are common in *E. albertii*, but several novel subtypes have also been identified in *E. albertii*, implying the pathogenic difference between *E. coli* and *E. albertii* [6,13]. The CDT is encoded by the *cdtABC* genes which were widely distributed in *E. albertii* [1]. The *cdtB* gene has been divided into six subtypes (*cdtB*-I to *cdtB*-VI), with *cdtB*-II and *cdtB*-VI being the most common subtypes [6,13]. Recently, several *cdtB*-II-positive *E. coli* isolates were reclassified as *E. albertii*, suggesting that previously identified *cdtB*-II-positive *E. coli* isolates might be *E. albertii*. [47]. *E. albertii cdtB*-II gene (*Eacdt*) was used to develop a PCR assay for the detection of *E. albertii* [20]. In this study, all isolates possessed *eae* and *cdtB*, but none carried *stx2*. The predominant *eae* and *cdtB* subtypes were sigma and *cdtB*-II, which were found to be common in clinical strains of *E. albertii* (Appendix A). Moreover, heat-stable enterotoxin gene *astA*, an important virulence gene in diarrheagenic *E. coli* [48], was also presented in several isolates of migratory birds. These indicated that the migratory bird-derived isolates may have pathogenic potential for humans.

The *E. albertii* isolates from migratory birds were classified into different phylogenetic clusters, indicating the genomic diversity of *E. albertii* in migratory birds in Poyang Lake. Several genotypes of *E. albertii* coexisted in a single individual, similar to other findings in raccoons and other birds [44,49]. Isolates belonging to the same clone were identified in two sampling sites (A and B) about 15 km apart, indicating clonal transmission in Poyang Lake.

Several gastroenteritis outbreaks caused by *E. albertii* have been reported [3]. Environmental water and vegetables were identified as some of the transmission vehicles in previous outbreaks [3,50]. Migratory birds are known to be involved in the maintenance and dissemination of zoonotic pathogens such as viruses, tick-borne pathogens, *Vibrio*, *Listeria monocytogenes*, *Salmonella enterica*, *Escherichia coli*, *Campylobacter jejuni*, and *Mycobacterium avium* [15,51,52,53]. These pathogens can also be transmitted to humans, animals, and poultry by contaminated water. Humans may come in contact with contaminated water for household or agricultural purposes. Therefore, migratory birds with a higher occurrence rate of *E. albertii* might contaminate environmental water, leading to human infections or outbreaks. In the present study, the isolates from migratory birds were genetically related to those isolated from human, poultry, bird, and dog from different regions or countries. Human, animal, and poultry might be infected by contact with contaminated water, soil, and food. The strains with an average of 101 cgSNPs indicated a highly close relationship with each other. Though the estimated number of cgSNPs per year for *E. albertii* or *E. coli* was unclear, a cutoff of ≤21 cgSNPs per genome per year for *Klebsiella pneumoniae* has been proposed [54]. These isolates with 101 cgSNP differences might share a recent common ancestor and clonal transmission. These data further proved that migratory birds might play a significant role in pathogen dissemination.

Our proposed method of bacterial isolation and identification faces several key limitations that must be acknowledged. First, the *E. albertii* specific primers proposed by Lindsey et al. [19] targeting *EAKF1_ch4033* can only correctly identify 96.5% (305/316) and missed 11 strains [9]. Second, the selection of white or colorless colonies in MacConkey agar might result in the exclusion of other lactose-fermenting *E. albertii* isolates [9]. These limitations might result in underestimation of the actual pathogen occurrence rate. We additionally acknowledge the limited sample size in this study; further large-scale epidemiological and more in-depth studies on migratory birds, other animal species, the environment and humans around Poyang Lake region are highly warranted to understand the significance of birds, and the transmission potential to the environment and humans due to *E. albertii* being carried.

In conclusion, this study proved that migratory birds may serve as an important reservoir of heterogeneous *E. albertii* with potential transmission sources to cause human infections. Considering the limited number of samples in this study, in the future, integrated, global, and ‘One Health’ approaches are critically needed to study *E. albertii*, an emerging zoonotic bacterial pathogen important in public health.

## Figures and Tables

**Figure 1 pathogens-12-00009-f001:**
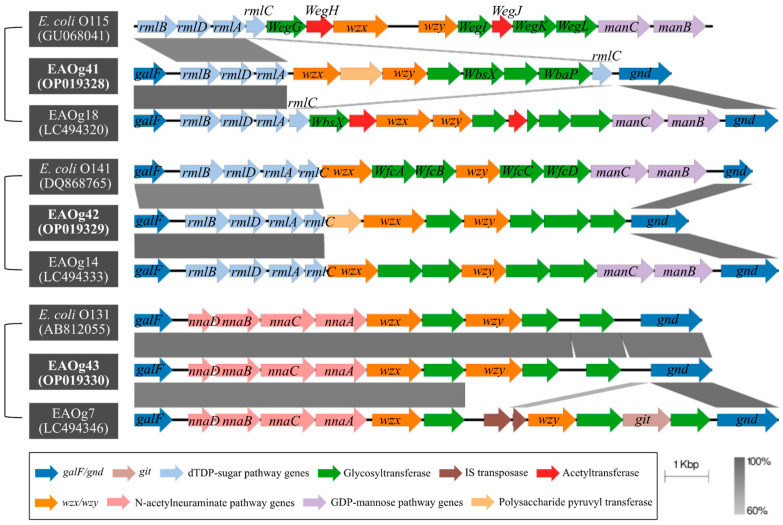
Comparison of the novel O-antigen biosynthesis gene clusters (O-AGCs) with homologous O-AGCs of known *E. coli* and *E. albertii.* The CDSs are labeled by arrows and colored by functional categories as shown in the boxed key. Gene names are displayed on arrows in bold italics. Grey shading indicates nucleotide identity between sequences according to BLASTn (60 to 100%). The regions outside the shaded regions represent no homology between genes. The outer scale is marked in kilobases. The reference strains of EAOg41, EAOg42, and EAOg43 were strains 104_2_TS_A, 222_1_EW_B, and 253_2_EW_B, respectively.

**Figure 2 pathogens-12-00009-f002:**
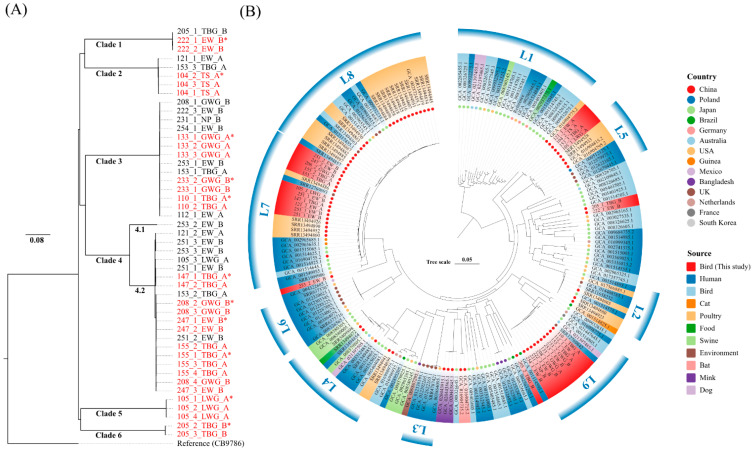
(**A**) A single nucleotide polymorphism (SNPs)-based phylogenetic tree of 47 *E. albertii* isolates in this study. The scale represents the evolutionary distances. Isolate details are summarized in their names: Sample ID_Number_Abbreviation of migratory birds_Sampling sites. The name with same sample ID in red color indicates these isolates could be the same clone from one sample. The asterisk symbol represents the representative strain selected for further genomic analysis. (**B**) An SNP-based phylogenetic tree of 193 *E. albertii* genome sequences. From inside to outside, the ring symbol shows the isolated countries, the sources, and the lineages. The leaves of tree were annotated with strain names (this study) or accession numbers of reference strains downloaded from NCBI or Enterobase.

**Figure 3 pathogens-12-00009-f003:**
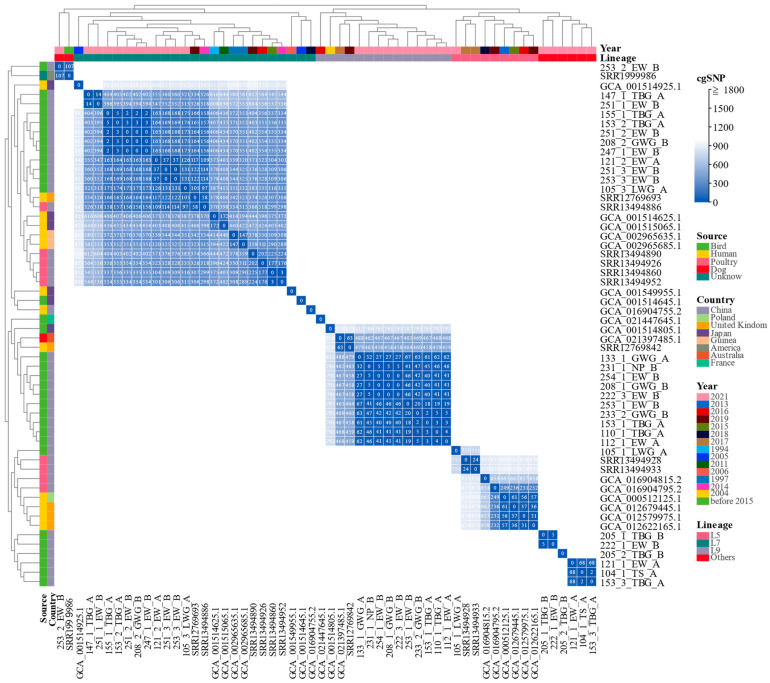
The heatmap of pairwise core genome SNP (cgSNP) values based on 53 strains including 29 migratory bird-derived isolates and 24 phylogenetically related strains. The color histogram shows the distribution of pairwise cgSNP values in the whole matrix, the stronger the blue zones, the more the strains are related to each other. The detailed matrix of cgSNPs < 500 was annotated with white words on the cell box. Color codes show the strain classification for year, lineage, country, and source.

**Table 1 pathogens-12-00009-t001:** Prevalence of *E. albertii* in different species of migratory birds.

Migratory Birds	No. of			
	Samples	PCR Positive (%)	Culture Positive (%)	Isolates
Eurasian wigeon (*Mareca penelope*)	31	8 (25.8%)	7 (22.6%)	16
Taiga bean goose (*Anser fabalis*)	19	7 (36.8%)	5 (26.3%)	14
Greater white-fronted goose (*Anser albifrons*)	11	5 (45.5%)	3 (27.3%)	9
Northern pintail (*Anas acuta*)	11	2 (18.2%)	1 (9.0%)	1
Swan goose (*Anser cygnoid*)	3	0 (0.0%)	0 (0.0%)	0
Lesser white-fronted goose (*Anser erythropus*)	2	1 (50.0%)	1 (50.0%)	4
Tundra swan (*Cygnus columbianus*)	2	1 (50.0%)	1 (50.0%)	3
Pied avocet (*Recurvirostra avosetta*)	1	0 (0.0%)	0 (0.0%)	0
Spotted redshank (*Tringa erythropus*)	1	0 (0.0%)	0 (0.0%)	0
Total	81	24 (29.6%)	18 (22.2%)	47

**Table 2 pathogens-12-00009-t002:** Characteristics of *E. albertii* isolates recovered from migratory birds in this study.

Isolate Name	Sample ID	Sampling Site ^a^	Source ^b^	*eae* Subtype	*cdtB* Subtype	O-AGCs Genotype	H-AGCs Genotype	Accession Number
104_1_TS_A	104	A	TS	epsilon3	II	EAOg41 *	EAHg4	JAMXMZ000000000
104_2_TS_A	104	A	TS	epsilon3	II	EAOg41 *	EAHg4	CP099914
104_3_TS_A	104	A	TS	epsilon3	II	EAOg41 *	EAHg4	JAMXNA000000000
105_1_LWG_A	105	A	LWG	nu	VI	EAOg6	EAHg1	CP099912-CP099913
105_2_LWG_A	105	A	LWG	nu	VI	EAOg6	EAHg1	JAMXNB000000000
105_3_LWG_A	105	A	LWG	sigma	II	EAOg2	EAHg4	JAMXNC000000000
105_4_LWG_A	105	A	LWG	nu	VI	EAOg6	EAHg1	JAMXND000000000
110_1_TBG_A	110	A	TBG	epsilon4	VI	EAOg21	EAHg3	CP099910-CP099911
110_2_TBG_A	110	A	TBG	epsilon4	VI	EAOg21	EAHg3	JAMXNE000000000
112_1_EW_A	112	A	EW	epsilon4	VI	EAOg21	EAHg3	CP099907-CP099909
121_1_EW_A	121	A	EW	epsilon3	II	EAOg41 *	EAHg4	CP099906
121_2_EW_A	121	A	EW	sigma	II	EAOg2	EAHg4	JAMXNF000000000
133_1_GWG_A	133	A	GWG	epsilon4	VI	EAOg21	EAHg3	CP099903-CP099905
133_2_GWG_A	133	A	GWG	epsilon4	VI	EAOg21	EAHg3	JAMXNG000000000
133_3_GWG_A	133	A	GWG	epsilon4	VI	EAOg21	EAHg3	JAMXNH000000000
147_1_TBG_A	147	A	TBG	sigma	II	EAOg1	EAHg4	CP099898-CP099902
147_2_TBG_A	147	A	TBG	sigma	II	EAOg1	EAHg4	JAMXNI000000000
153_1_TBG_A	153	A	TBG	epsilon4	VI	EAOg21	EAHg3	JAMXNJ000000000
153_2_TBG_A	153	A	TBG	sigma	II	EAOg2	EAHg4	CP099895-CP099897
153_3_TBG_A	153	A	TBG	epsilon3	II	EAOg41 *	EAHg4	JAMXNK000000000
155_1_TBG_A	155	A	TBG	sigma	II	EAOg2	EAHg4	JAMXNL000000000
155_2_TBG_A	155	A	TBG	sigma	II	EAOg2	EAHg4	CP099892-CP099894
155_3_TBG_A	155	A	TBG	sigma	II	EAOg2	EAHg4	JAMXNM000000000
155_4_TBG_A	155	A	TBG	sigma	II	EAOg2	EAHg4	JAMXNN000000000
205_1_TBG_B	205	B	TBG	epsilon1	II	EAOg42 *	EAHg3	JAMXNO000000000
205_2_TBG_B	205	B	TBG	N3	VI	EAOg8	EAHg1	CP099890-CP099891
205_3_TBG_B	205	B	TBG	N3	VI	EAOg8	EAHg1	JAMXNP000000000
208_1_GWG_B	208	B	GWG	epsilon4	VI	EAOg21	EAHg3	JAMXNQ000000000
208_2_GWG_B	208	B	GWG	sigma	II	EAOg2	EAHg4	CP099887-CP099889
208_3_GWG_B	208	B	GWG	sigma	II	EAOg2	EAHg4	JAMXNR000000000
208_4_GWG_B	208	B	GWG	sigma	II	EAOg2	EAHg4	JAMXNS000000000
222_1_EW_B	222	B	EW	epsilon1	II	EAOg42 *	EAHg3	CP099886
222_2_EW_B	222	B	EW	epsilon1	II	EAOg42 *	EAHg3	JAMXNT000000000
222_3_EW_B	222	B	EW	epsilon4	VI	EAOg21	EAHg3	JAMXNU000000000
231_1_NP_B	231	B	NP	epsilon4	VI	EAOg21	EAHg3	CP099883-CP099885
233_1_GWG_B	233	B	GWG	epsilon4	VI	EAOg21	EAHg3	JAMXNV000000000
233_2_GWG_B	233	B	GWG	epsilon4	VI	EAOg21	EAHg3	CP099880-CP099882
247_1_EW_B	247	B	EW	sigma	II	EAOg2	EAHg4	JAMXNW000000000
247_2_EW_B	247	B	EW	sigma	II	EAOg2	EAHg4	CP099877-CP099879
247_3_EW_B	247	B	EW	sigma	II	EAOg2	EAHg4	JAMXNX000000000
251_1_EW_B	251	B	EW	sigma	II	EAOg1	EAHg4	JAMXNY000000000
251_2_EW_B	251	B	EW	sigma	II	EAOg2	EAHg4	CP099874-CP099876
251_3_EW_B	251	B	EW	sigma	II	EAOg2	EAHg4	JAMXNZ000000000
253_1_EW_B	253	B	EW	epsilon4	VI	EAOg21	EAHg3	JAMXOA000000000
253_2_EW_B	253	B	EW	epsilon3	II	EAOg43 *	EAHg1	CP099871-CP099873
253_3_EW_B	253	B	EW	sigma	II	EAOg2	EAHg4	JAMXOB000000000
254_1_EW_B	254	B	EW	epsilon4	VI	EAOg21	EAHg3	CP099868-CP099870

* The novel O-AGC genotypes identified in this study. ^a^ Site A: 116°11′35.178″ N, 29°15′15.422″ E; Site B: 116°18′38.912″ N, 29°9′57.287″ E. ^b^ Sources: TS, Tundra swan (*Cygnus columbianus*); LWG, Lesser white-fronted goose (*Anser erythropus*); TBG, Taiga bean goose (*Anser fabalis*); EW, Eurasian wigeon (*Mareca penelope*); GWG, Greater white-fronted goose (*Anser albifrons*).

## Data Availability

All genomes in this work were submitted to GenBank under the accession numbers CP099868–CP099914 and JAMXMZ000000000–JAMXOB000000000. The annotated sequences of O-antigen biosynthesis gene clusters were submitted to GenBank under accession numbers OP019328-OP019330.

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
