# Peer review of "Identification and Genomic Characterization of Escherichia albertii in Migratory Birds from Poyang Lake, China"

_pathogens, 2022, doi:10.3390/pathogens12010009_

Round 1

Reviewer 1 Report

Overview: The paper investigates the role of migratory birds as reservoirs and vehicles for the transmission of the emerging zoonotic foodborne enteropathogen Escherichia albertii. A fecal contamination rate of 29.6% was observed as determined by PCR. Isolates were recovered from 18 samples and were characterized through WGS revealing 3 novel EAOgs. Some of these isolates were phylogenetically closely related to previous isolates from humans and poultry suggesting that migratory birds may be an E. albertii reservoir playing a significant role in the transmission cycles of this pathogen.

The study presents important data that will aid in understanding E. albertii epidemiology, host range, and prevalence. However, major revisions of the manuscript are needed.

Comments

Abstract and Introduction

Line 18: Please check grammar, remove “were”

Line 32: Please check grammar, remove “then” 

Line 33: Please replace “specific” with “distinguishing”

Methods and Results

Line 66: breeding areas

Line 87-: Please note that for lysP and mdh, some studies (Hinenoya et al., 2019b, Maeda et al., 2014) have reported nonspecific amplification in assays targeting these genes indicative of lower sensitivity for such primers. Please discuss this study limitation.

Ref: Maeda E, Murakami K, Okamoto F, Etoh Y, Sera N, Ito K, Fujimoto S. Nonspecificity of primers for Escherichia albertii detection. Jpn J Infect Dis. 2014;67(6):503-5. doi: 10.7883/yoken.67.503.

Line 88-89: Are the four isolates derived from this process different strains or different isolates of the same strain? A look at your results suggests that most of these 4 isolates per sample might be the same strain. Reporting isolates of the same strain as different strains will result in false representation, misleading the population structure and the prevalence of virulence genes in E. albertii.

Line 91-103: What influenced the antibiotic choices used in the study?

Line 105: Which media was used?

Line 105 and 116: Please determine if the four isolates derived from the same samples are different strains or multiple isolates of the same strain.

Line 138. Line 224-234: This analysis needs to be revised. The authors sequenced and analyzed 47 strains isolated in this study. However, some isolates (e.g., 104_1_TS_A, 104_2_TS_A, and 104_3_TS_A, isolates 105_1_LWG_A, 105_2_LWG_A, and 105_4_LWG_A, isolates 110_1_TBG_A and 110_2_TBG_A, and isolates 155_1_TBG_A, 155_2_TBG_A, 155_3_TBG_A, and 155_4_TBG_A), which have been isolated from the same bird and with no or very few SNPs, look like the same strains, i.e. different isolates of the same strain. I recommend removing these duplicate isolates from the analyses to prevent misleading the population structure and the prevalence of virulence genes in E. albertii.

Discussion:

The discussion must highlight the importance of E. albertii environmental contamination linking it to known associated problems e.g., outbreaks.

Line 290-291: Is serotype (O and H genotype) associated with any traits e.g., virulence in E. albertii. I would add a reference here.

Line 296: Please check grammar and rephrase.

Line 314: Several?

Line 315-316: I would be cautious with this statement, as it seems that the four isolates often obtained per sample were the same strain.

Line 318-320: Please check grammar and rephrase.

Line 322-324: Please check grammar and rephrase. Please add more references here.

Line 327: Please check grammar and rephrase.

Tables and Figures:

Table 2 (Line 194) and Figure 3 (Line 273): There are 0-1 SNP differences between isolates 104_1_TS_A, 104_2_TS_A, and 104_3_TS_A, isolates 105_1_LWG_A, 105_2_LWG_A, and 105_4_LWG_A, isolates 110_1_TBG_A and 110_2_TBG_A, and isolates 155_1_TBG_A, 155_2_TBG_A, 155_3_TBG_A, and 155_4_TBG_A.

These isolates have the same virulence gene and antibiotic sensitivity profiles as well as O and H genotypes (Table 2, Table S1, and Table S2). They also cluster together (Figure 2) suggesting that they are isolates of the same strain and not different strains as currently presented in the manuscript. I recommend removing these duplicate isolates from the analyses to prevent misleading the population structure and virulence-related genes prevalence in E. albertii.

Reviewer 2 Report

The authors described the genomic characterization of Escherichia albertii in migratory birds. The study has been written well and scientifically sound, except for a few minor corrections to improve the manuscript's clarity.

1.     Line 80: The full name of the 'EC broth" should be "Escherichia coli broth" when mentioned first.

2.     Line 90: It is suggested to mention the ATCC control strains used to ensure the results of antimicrobial susceptibility testing under the heading "Antimicrobial susceptibility test."

3.     It is suggested to replace some older references in the discussion section with newly updated citations.

Round 2

Reviewer 1 Report

Thank you for the much-improved manuscript.

Line 22: Please check grammar. I would remove "of".

Line 24: Please check grammar. I would remove "and". Suggested: ", thereby acting as potential...".

Line 311-312: This statement is not clear, please revise it. Also, check your grammar here. 

Line 325: Please check grammar and rephrase. 

Line 340: Please rephrase from "as one of" to "as some of"

Line 344: also

Line 360-361: Please add supporting references here. 
